# SUMO-Activating Enzyme Subunit 1 (SAE1) Is a Promising Diagnostic Cancer Metabolism Biomarker of Hepatocellular Carcinoma

**DOI:** 10.3390/cells10010178

**Published:** 2021-01-17

**Authors:** Jiann Ruey Ong, Oluwaseun Adebayo Bamodu, Nguyen Viet Khang, Yen-Kuang Lin, Chi-Tai Yeh, Wei-Hwa Lee, Yih-Giun Cherng

**Affiliations:** 1Department of Emergency Medicine, Taipei Medical University—Shuang Ho Hospital, New Taipei City 235, Taiwan; 12642@s.tmu.edu.tw (J.R.O.); 19567@s.tmu.edu.tw (N.V.K.); 2Graduate Institute of Injury Prevention and Control, Taipei Medical University, Taipei 110, Taiwan; 3Department of Emergency Medicine, School of Medicine, Taipei Medical University, Taipei 110, Taiwan; 4Department of Medical Research & Education, Taipei Medical University—Shuang Ho Hospital, New Taipei City 235, Taiwan; 16625@s.tmu.edu.tw (O.A.B.); 19567@s.tmu.edu.tw (N.V.K.); robbinlin@tmu.edu.tw (Y.-K.L.); ctyeh@s.tmu.edu.tw (C.-T.Y.); 5Department of Medical Laboratory Science and Biotechnology, Yuanpei University of Medical Technology, Hsinchu 300, Taiwan; 6Department of Pathology, Taipei Medical University—Shuang Ho Hospital, New Taipei City 235, Taiwan; whlpath97616@s.tmu.edu.tw; 7Department of Anesthesiology, Shuang Ho Hospital, Taipei Medical University, New Taipei City 235, Taiwan; 8Department of Anesthesiology, School of Medicine, College of Medicine, Taipei Medical University, Taipei 110, Taiwan

**Keywords:** SAE1, hepatocellular carcinoma, SUMOylation, diagnosis, prognosis, metastasis

## Abstract

Hepatocellular carcinoma (HCC) is one of the most diagnosed malignancies and a leading cause of cancer-related mortality globally. This is exacerbated by its highly aggressive phenotype, and limitation in early diagnosis and effective therapies. The SUMO-activating enzyme subunit 1 (SAE1) is a component of a heterodimeric small ubiquitin-related modifier that plays a vital role in SUMOylation, a post-translational modification involving in cellular events such as regulation of transcription, cell cycle and apoptosis. Reported overexpression of *SAE1* in glioma in a stage-dependent manner suggests it has a probable role in cancer initiation and progression. In this study, hypothesizing that *SAE1* is implicated in HCC metastatic phenotype and poor prognosis, we analyzed the expression of *SAE1* in several cancer databases and to unravel the underlying molecular mechanism of SAE1-associated hepatocarcinogenesis. Here, we demonstrated that *SAE1* is over-expressed in HCC samples compared to normal liver tissue, and this observed *SAE1* overexpression is stage and grade-dependent and associated with poor survival. The receiver operating characteristic analysis of *SAE1* in TCGA−LIHC patients (*n* = 421) showed an AUC of 0.925, indicating an excellent diagnostic value of *SAE1* in HCC. Our protein-protein interaction analysis for SAE1 showed that SAE1 interacted with and activated oncogenes such as *PLK1*, *CCNB1*, *CDK4* and *CDK1*, while simultaneously inhibiting tumor suppressors including *PDK4*, *KLF9*, *FOXO1* and *ALDH2*. Immunohistochemical staining and clinicopathological correlate analysis of SAE1 in our TMU-SHH HCC cohort (*n* = 54) further validated the overexpression of SAE1 in cancerous liver tissues compared with ‘normal’ paracancerous tissue, and high SAE1 expression was strongly correlated with metastasis and disease progression. The oncogenic effect of upregulated *SAE1* is associated with dysregulated cancer metabolic signaling. In conclusion, the present study demonstrates that SAE1 is a targetable cancer metabolic biomarker with high potential diagnostic and prognostic implications for patients with HCC.

## 1. Introduction

Hepatocellular carcinoma (HCC) is the fifth most commonly diagnosed cancer and ranks as the third commonest cause of cancer-related mortality, accounting for more than 700,000 fatalities in the world, annually [1]. The major risk factors for HCC include chronic infection of hepatitis B and C viruses (HBV and HCV), cirrhosis, alcohol abuse and non-alcoholic fatty liver disease (NAFLD) [2]. Hepatocarcinogenesis is characterized by dysregulated activation and/or expression of relevant genes in/on the hepatocytes, with resultant oncogene upregulation and tumor suppressor downregulation [3]. The last 5 decades has been characterized by discovery several biomarkers for diagnosis of HCC, including the α-fetoprotein (AFP), AFP-L3 (a heteroplast of AFP), des-γ-carboxyprothrombin (DCP), α-L-fucosidase (AFU), golgi protein 73 (GP73), osteopontin (OPN) and carbohydrate antigen 19-9 (CA19-9), which is globally regarded as diagnostic serological biomarkers for diagnosis of HCC patients. However, due to the clonal evolution, intratumoral and interpatient heterogeneity of HCC [4,5], like AFP, the diagnostic validity and clinical applicability of all these serological biomarkers remain debatable, especially considering their sub-optimal diagnostic specificity and sensitivity for early detection of HCC [6,7].

Similarly, histochemical biomarkers of HCC including glypican-3 (GPC-3), hepatocyte paraffin 1 (Hep Par 1), heat shock protein 70 (HSP70), glutamine synthetase (GS), arginase-1 (Arg-1), cytokeratin 7 and 19 (CK7 and CK19) are also plagued with same weakness in spite of their overall strength [8,9]. Against the background of this diagnostic challenge, the discovery of a biomarker with high and reliable diagnostic and prognostic accuracy and validity remain an unmet need in hepato-oncology clinics. Thus, the exploration for such biomarker in the present study; with the ultimate aim of proffering a therapeutic target, as well as improving the accuracy of diagnosis and efficacy of treatment modality in patients with HCC.

The disease course and progression of HCC is facilitated by altered cellular gene expression with dysregulated metabolism and pathophysiological signaling pathways [3,4,5]. SUMOylation, a post-translational modification that entails addition of small ubiquitin-like modifier (SUMO) groups to target proteins, is involved in numerous cellular events including transcriptional regulation, protein stability, cell cycle and apoptosis [10]. Upregulated expression of *SAE1* (SUMO-activating enzyme subunit 1), an essential heterodimeric SUMO-activating effector of SUMOylation, has been implicated in the tumorigenesis and progression of several human malignancies, including in glioma [11], gastric cancer [12] and, more broadly, in Myc-driven carcinomas [13,14]; however, the biological roles of SAE1 in HCC remains underexplored.

In the present study, hypothesizing that *SAE1* is implicated in HCC metastatic phenotype and poor prognosis, we investigated the variability of SAE1 expression in several cancer databases and its probable implication in HCC progression. Results presented herein indicate that, compared to normal liver samples, SAE1 is overexpressed in HCC, associated with the enhanced metastatic phenotype, disease progression, and poor prognosis of patients with HCC, thus indicating that SAE1 possesses reliable and clinically-relevant diagnostic value and is a potential novel biomarker of prognosis for HCC.

## 2. Materials and Methods

### 2.1. HCC Samples and Cohort Characterization

Clinical samples of patients with HCC were retrieved from the HCC tissue archive of the Taipei Medical University—Shuang Ho Hospital (TMU-SHH), New Taipei, Taiwan. After exclusion of cases with incomplete clinical information and insufficient sample for biomedical assays, only 54 clinical samples were used in the present study. This study was approved by the Institutional Human Research Ethics Review Board (TMU-JIRB No. 201302016) of Taipei Medical University.

### 2.2. Data Acquisition and Statistical Analysis of HCC

The raw gene expression data of *SAE1* and related genes obtained by RNA sequencing (RNA-seq) along with clinical data were downloaded from the freely-accessible Genotype-Tissue Expression (GTEx) (https://gtexportal.org/), The Cancer Genome Atlas (TCGA) (https://xenabrowser.net/) and the Gene Expression Omnibus (GEO) (https://www.ncbi.nlm.nih.gov/geo/) databases. All data were visualized and analyzed using the GraphPad Prism version 8.0.0 for Windows, (GraphPad Software, San Diego, California USA, www.graphpad.com). Hazard ratios obtained from the analysis of overall and progression-free survival curves in various TCGA databases were visualized using forest plots. STRING version 11.0 (https://string-db.org/) was used for visualization of protein-protein interaction network and functional enrichment analysis.

### 2.3. Immunohistochemistry

Standard immunohistochemical (IHC) staining and the quantitation of the staining were performed as previously described [15]. Briefly, after de-waxing of the 5μm thick sections using xylene and re-hydration with ethanol, endogenous peroxidase activity was blocked using 3% hydrogen peroxide. This was followed by antigen retrieval, blocking with 10% normal serum, and incubation of the sections with anti-SAE1 (1:500; #ab185552, Abcam, Cambridge, UK), anti-SUMO1 (1:100; #ab32058, Abcam), anti-SUMO2 (1:100; #ab212838, Abcam), and UBC9 (1:100; #ab75854, Abcam) antibodies overnight at 4 °C, followed by goat anti-rabbit IgG (H + L) HRP-conjugated secondary antibody (1:10,000; #65-6120, Thermo Fisher Scientific Inc., Waltham, MA, USA). As chromogenic substrate, Diaminobenzidine (DAB) was used, and the stained sections were counter-stained with Gill’s hematoxylin (Thermo Fisher Scientific, Waltham, MA, USA). The univariate and multivariate analyses were done using the Cox proportional hazards regression model.

### 2.4. SAE1 Knockdown Using CRISPR Interference

Plasmid vectors containing pLV hU6-sgRNA hUbC-dCas9-KRAB-T2a-Puro (Plasmid #71236) was used for SAE1 knockdown in cells by CRISPR interference (CRISPRi). Three SAE1-specific single-guide RNAs (sgRNAs) designed using the online tool CHOPCHOP (http://chopchop.cbu.uib.no/) were synthesized and separately cloned into lenti-dCas9-KRAB. Lentiviruses were packaged and transfected into Huh7 cells. Transfected monoclonal Huh7 cells were selected by 2 μg/mL puromycin. The cell construction with knockdown of SAE1 was verified by genomic sequencing and quantitative real-time PCR. The sgRNA sequences for SAE1 are as follows: sgSAE1#1 (sg#1) 5′-GTGCCACATAAGTG ACCACG-3′, sgSAE1#2 (sg#2) 5′-GGCGACTGCATGTCACGTGA-3′ and sgSAE1#3 (sg#3) 5′-ACGAGGTACT GCGCAGGCGT-3′.

### 2.5. Real-Time PCR Reaction

Quantitative real-time PCR reaction was performed as previous described in [15] using the following primers: SAE1-FP: 5′-AGGACTGACCATGCTGGATCAC-3′ and SAE1-RP: 5′-CTCAGTGTCC ACCTTCACATCC-3′. 

### 2.6. Western Blot Analysis

Total protein lysate was prepared from cultured HCC cells using ice-cold lysis buffer solution. After boiling at 95 °C for 5 min, immunoblotting was performed. Blots were blocked with 5% non-fat milk in Tris Buffered Saline with Tween 20 (TBST) for 1 h, incubated at 4 °C overnight with specific primary antibodies against SAE1 (1:1000; #13585S, Cell Signaling Technology, Inc., Danvers, MA, USA), CDK4 (1:1000; #2906, Cell Signaling Technology), Cyclin B1 (1:500; Sc-245, Santa Cruz Biotechnology, Dallas, TX, USA), FOXO1 (1:1000; #2880, Cell Signaling Technology), GAPDH (1:500; Sc-47724, Santa Cruz Biotechnology), and KLF9 (1:1000; ab227920, Abcam, Cambridge Inc., Cambridge, UK) in Appendix A. Thereafter, the polyvinylidene difluoride (PVDF) membranes were washed thrice with TBST, incubated with horseradish peroxidase (HRP)-labeled secondary antibody for 1 h at room temperature and then washed with TBST again before band detection using enhanced chemiluminescence (ECL) Western blotting reagents and imaging with the BioSpectrum Imaging System (UVP, Upland, CA, USA).

### 2.7. Transwell Matrigel Invasion Assay

After pre-coating the chamber membranes (8 μm, BD Falcon) with Bmatrigel at 4 °C overnight, the wild type (WT) or CRISPRi SAE1-knockdown cells were seeded at a density of 1 × 10^5^ cells per chamber. DMEM with 1% fetal bovine serum (FBS) supplement was added to the upper chamber and DMEM containing 10% FBS added to the lower chamber. Cells were incubated for 48 h. The non-invading cells on the top of membranes was carefully removed using sterile cotton swab, and the invaded cells that penetrate the membrane were fixed in ethanol, followed by crystal violet staining. The number of invaded cells was counted under the microscope in five random fields of vision and representative images photographed.

### 2.8. Scratch-Wound Migration Assay

Cell migration potential was evaluated using the wound healing assay. Briefly, wild type (WT) or CRISPRi SAE1-knockdown Huh7 cells were seeded onto 6-well plates (1 × 10^6^ cells/well) (Corning Inc., Corning, NY, USA) with complete growth media containing 0.2% FBS, and cultured till 95–100% confluence was attained. That would help cells not going to apoptosis or necrosis, but also no proliferation occurs. The cell monolayers were scratched with sterile yellow pipette tip along the median axes of the culture wells. The cell migration images were captured at the 0 and 16 h time points after denudation, under a microscope with a 10× objective lens, and analyzed with the NIH ImageJ software v1.49 (https://imagej.nih.gov/ij/download.html).

### 2.9. Statistical Analysis

All assays were performed at least thrice in triplicate. Values are expressed as the mean ± standard deviation (SD). Comparisons between groups were estimated using Student’s t-test for cell line experiments or the Mann–Whitney U-test for clinical data, Spearman’s rank correlation between variables, and the Kruskal–Wallis test for comparison of three or more groups. The Kaplan–Meier method was used for the survival analysis, and the difference between survival curves was tested by a log-rank test. Univariate and multivariate analyses were based on the Cox proportional hazards regression model. All statistical analyses were performed using IBM SPSS Statistics for Windows, version 20 (IBM, Armonk, NY, USA). A *p*-value < 0.05 was considered statistically significant.

## 3. Results

### 3.1. Gene Expression Profile of SAE1 in Pan-Cancer Cohort

To examine the expression of *SAE1* in various tissue types, we analyzed expression data of samples (*n* = 17,382) derived from non-disease tissues (*n* = 54) obtained from 948 donors using the Genotype-Tissue Expression (GTEx) project (GTEx Analysis Release V8 (dbGaP Accession phs000424.v8.p2) [16]. The lowest expression of *SAE1* was observed in liver (*n* = 110), pancreas (*n* = 167), kidney (*n* = 27) and pituitary (*n* = 107), in increasing order of magnitude, while testis (*n* = 165) and bone marrow (*n* = 70) exhibited the highest *SAE1* expression levels (Figure 1A).

Further exploring the *SAE1* mRNA levels in paired tumor-non-tumor samples from patients with one of 18 different cancer types using The Cancer Genome Atlas (TCGA) datasets, we observed that *SAE1* was significantly more expressed in liver hepatocellular carcinoma (LIHC, *n* = 371), compared to their normal tissue counterparts (*n* = 50) (Figure 1B). The upregulation of *SAE1* expression was also found in several other cancer types, including lung squamous cell carcinoma (LUSC, *n* = 553), colon adenocarcinoma (COAD, *n* = 327), head and neck squamous cell carcinoma (HNSC, *n* = 534), kidney chromophobe (KICH, *n* = 91), breast invasive carcinoma (BRCA, *n* = 1211), cervical squamous cell carcinoma and endocervical adenocarcinoma (CESC, *n* = 306) and uterine corpus endometrial carcinoma (UCEC, *n* = 211) (Figure 1C).

### 3.2. SAE1 Is Overexpressed in HCC and Associated with Disease Progression

Having demonstrated that *SAE1* is significantly more expressed in HCC compared with the non-tumor samples (~1.1-fold, *p* < 0.0001) (Figure 2A), to minimize probable experimental design-based bias, we excluded unpaired cases (*n* = 321) and analyzed the expression of *SAE1* in only cases with paired tumor–non-tumor samples (*n* = 100).

Our results indicate that regardless of excluded cases, the median expression of *SAE1* mRNA remained significantly upregulated in the tumor samples (~1.1-fold, *p* < 0.0001) (Figure 2B). Probing for probable role of *SAE1* in disease progression, we demonstrated that *SAE1* expression increased with HCC stage, as evidenced by higher expression in advanced stages (stages III/IV) and stage II than in stage I or non-tumor (stages II-IV > stage I >> non-tumor) (*p* < 0.0001), indicating the increased expression of *SAE1* is tumorigenic and disease progression-associated (Figure 2C). Supporting the results above, our analysis of four other HCC cohort datasets downloaded from the Gene Expression Omnibus (GEO): GSE36376 (Park 2012, *n* = 433), GSE64041 (Makowska 2014, *n* = 120), GSE14520 (Wang 2009, *n* = 445), GSE76297 (Wang 2015, *n* = 304) showed that *SAE1* was significantly overexpressed in all these four datasets (Figure 2D), further confirming the overexpression of the gene in TCGA−LIHC. Concordantly, we demonstrated that the expression of *SAE1* increased as histologic grade increased (*p* < 0.0001), was equivocal for gender, and mildly higher in patients aged <60 (Appendix A). More so, *SAE1* expression in T1 < T2 < T3 < T4 (*p* = 0.0009), mildly higher in N1 and M1 compared with N0 (*p* = 0.255) and M0 (*p* = 0.682), respectively (Appendix A). Expectedly, patients with residual tumor (R1 and R2) has higher expression of *SAE1* compared with R0 (*p* = 0.958), and statistically significant upregulation of *SAE1* was observed in the deceased compared to those alive (*p* = 0.025) and equivocal for radiation therapy (Appendix A). These results indicate the overexpression of *SAE1* in HCC, and its association with disease progression in a stage- and grade-dependent manner.

### 3.3. The Overexpression of SAE1 Is Associated with Metastasis and Poor Prognosis in Patients with HCC

The eligible subjects in the TMU-SHH HCC cohort were aged from 25 to 85 with median age of 58.24 for patients with high SAE1 (*n* = 25) and 61.14 for those with low SAE1 (*n* = 29). 9 (16.67%) were male and 45 (83.33%) were female. Analyzed clinicopathological data, including patients’ demographic (age and gender) and biochemical profile (AFP, lymph node metastasis, tumor stages and survival status) are summarized in Table 1. The cut-off for AFP is based on clinical consensus that a significantly high level of circulating AFP, greater than 400 ng/mL is suggestive of malignancy of the liver. On the other hand, a threshold cut-off of 350 was ascribed for differential expression of SAE1 based on the quick (Q) score derived from the staining intensity and distribution of SAE1 in the clinical samples.

Furthermore, employed the Cox proportional hazard model for clinicopathological analysis of SAE1 protein expression, along with disease-specific risk factors, including age, gender, AFP and metastasis in the TMU-SHH HCC cohort (*n* = 54). Results of both univariate and multivariate analyses revealed that high SAE1 protein expression level is strongly associated with metastasis (Table 2).

Corroborating the findings from big data analysis, IHC staining of samples from our TMU-SHH HCC cohort (*n* = 54) showed a 1.85-fold upregulated expression of SAE protein in the cancerous HCC compared to the non-tumor para-cancer liver tissue (*p* < 0.0001) (Figure 3A).

Probing for clinical relevance of the observed high expression of SAE1 protein, using the Kaplan-Meier curve for survival analysis, we demonstrated that compared to patients with low SAE1 expression (*n* = 20), those with high SAE1 expression (*n* = 18) exhibited worse overall survival ((HR (95%CI): 5.578 (1.250–24.890); *p* = 0.024)) (Figure 3B). Consistent with the vital role of SUMO proteins and the UBC9 in SUMOylation [17,18,19,20,21], we further demonstrated that similar to SAE1, the expression levels of SUMO1, SUMO2, and UBC9 were upregulated in the HCC tissues compared to their non-tumor paracancerous counterparts (Figure 3C). These results indicate that overexpression of SAE1, a critical component of the SUMOylation complex, is associated with metastasis and poor prognosis in patients with HCC.

### 3.4. SAE1 Is a Reliable Diagnostic and Prognostic Biomarker for HCC

To evaluate the diagnostic and prognostic validity of *SAE1* in HCC, we performed a survival analysis of the *SAE1* expression-stratified TCGA−LIHC dataset using the Kaplan-Meier plots and receiver operating characteristic (ROC) curves. We demonstrated that patients with high *SAE1* expression exhibited worse overall survival (OS) (HR = 1.873, *p* = 0.0004), disease-survival (DSS) (HR = 2.070, *p* = 0.0016), and progression-free survival (PFS) (HR = 1.809, *p* < 0.0001) over a follow-up period of 10 years (Figure 4A–C).

In addition, we found that patient with advanced stage HCC exhibited worse OS compared with those in early stage (stage III/IV vs. I/II: *p* < 0.0001) (Appendix A). Furthermore, from our intergroup analysis of *SAE1* expression in HCC vs. normal liver for diagnostic implication, the area under the ROC curve (AUC) was 0.925 (Youden’s J = 0.71, SE = 0.01, *p* < 0.0001) (Figure 4D), with hazard ratios and 95% confidence intervals of 1.873 (1.321–2.656) and 1.809 (1.345–2.434) for OS and PFS, respectively (Figure 4E,F). More so, compared with *SAE1* expression in non-tumor, the AUCs for *SAE1* expression in patients with stages I, II, III, and IV were 0.92 (*p* < 0.0001), 0.93 (*p* < 0.0001), 0.94 (*p* < 0.0001), and 1.00 (*p* = 0.0003), respectively (Appendix A).

### 3.5. SAE1 Upregulates Oncogenic Effectors of Cell Cycle Progression while Downregulating FOXO1-Associated Tumor Suppressing Signaling

To unravel the underlying molecular mechanism of already documented SAE1-associated hepatocarcinogenesis, we probed for genes concomitantly upregulated or suppressed when SAE1 is upregulated, and SAE1-dependent protein-protein interaction (PPI). Using the STRING database (https://string-db.org) for visualization of probable network of SAE1-associated functional proteins in humans, we found that SAE1 exhibited strong interaction with SUMO proteins such as SAE2 (also called UBA2), SUMO-conjugating enzyme E2I (UBE2I/UBC9) and SUMO specific peptidase 1 (SENP1), neural precursor cell-expressed developmentally down-regulated protein 8 (NEDD8), ubiquitin-conjugating enzyme E2M (UBE2M), RAN GTPase-activating protein 1 (RANGAP1), RAN binding protein 2 (RANBP2), RWD domain containing protein 3 (RWDD3), cullin-4A (CUL4A), cullin-5 (CUL5), cullin-associated NEDD8-dissociated protein 1 (CAND1), RING-box protein 1 (RBX1), S-phase kinase-associated protein 1/2 (SKP1/2), and defective in cullin neddylation 1 domain-containing 1 (DCUN1D1) protein (Figure 5A).

Furthermore, we used the cBioPortal for Cancer Genomics (https://www.cbioportal.org/) for the identification of genes with significant positive or negative correlation with SAE1 in the TCGA−LIHC cohort. Our results showed that *SAE1* is strongly co-expressed with the cell cycle-related oncogenes *PLK1* (*r* = 0.64, *p* < 0.0001), *CCNB1* (*r* = 0.64, *p* < 0.0001), *CDK4* (*r* = 0.58, *p* < 0.0001) and *CDK1* (*r* = 0.58, *p* < 0.0001) (Figure 5B), but inversely related to tumor suppressor genes *PDK4* (*r* = −0.47, *p* < 0.0001), *KLF9* (*r* = −0.47, *p* < 0.0001), *FOXO1* (*r* = −0.42, *p* < 0.0001) and *ALDH2* (*r* = −0.5131, *p* < 0.0001) (Figure 5C). To gain inside into the significance of SAE1 in hepatocarcinogenesis, we knocked down the gene using CRISPRi and validated the knockdown efficacy by real-time PCR analysis. As shown in Figure 5D, sgSAE1#1 and sgSAE1#3 exhibited high knockdown efficacy with roughly 50% and 30%, respectively. Consistent with these data, results of our western blot analysis show that silencing SAE1, elicited upregulated expression of drivers of cancer progression CDK4 and cyclin B1 (a *CCNB1* gene product), concomitantly with downregulated tumor suppressors FOXO1 and KLF9 proteins in HCC cell line Huh7 (Figure 5E). Similarly, sgSAE1#3 significantly suppressed the ability of the Huh7 cells to invade (5.2-fold, *p* < 0.001) or migrate (6.1-fold, *p* < 0.001) (Figure 5F). These findings suggest that SAE1 upregulates oncogenic effectors of cell cycle progression while downregulating FOXO1-associated tumor suppressing signaling.

### 3.6. The Oncogenic Effect of Upregulated SAE1 Is Associated with Dysregulated Cancer Metabolic Signalings

Understanding that several stress signals, including hypoxia, impaired metabolism, nutrient deficiency, DNA damage (genotoxic stress) and dysregulated nucleotide metabolism, facilitate the initiation and development of cancer, and that dysregulated SUMOylation can play a crucial role in the protection of cancer cells from exogenous or endogenous stress signals [21], we investigated likely association of SAE1 expression with hypoxia and impaired metabolism. The results of our gene set enrichment analysis of the LIHC (*n* = 371), GSE14520 (*n* = 225), GSE36376 (*n* = 240), and GSE64041 (*n* = 60) HCC datasets showed the existence of significant positive correlation between high SAE1 and dysregulated reactive oxygen species (ROS), glycolysis, and cholesterol homeostasis pathways in patients with HCC (Figure 6A). Gene Set Enrichment Analysis (GSEA) plots using HCC datasets implied the upregulated co-expression of *SAE1* and biomarkers of glucose metabolism, pyrimidine metabolism, and purine metabolism (Figure 6B). These data do indicate, at least in part, that the oncogenic effect of upregulated SAE1 is associated with dysregulated cancer metabolic signaling.

## 4. Discussion

Hepatocarcinogenesis entails alteration of cellular gene expression with consequent loss of benignity, acquisition of malignant phenotype and enhancement of the aggressiveness of the resultant cancerous liver cells [3]. Previous studies have shown that post-translational modification such as ubiquitination and SUMOylation, which play very crucial roles in the non-static regulation of protein structure, stability, intracellular localization, activity, function, and interaction with other proteins, are significantly enhanced in HCC [14,17]. Recently, it was reported that the SUMO2-mediated SUMOylation of the supposedly tumor suppressor, liver kinase B1 (LKB1), facilitated hepatocarcinogenesis and disease progression in in vivo mice models and human HCC cohort, especially in hypoxic conditions [17]. Consistent with this report, it has also been shown that hypoxia or exposure to TNF-α upregulated SUMO1 expression and the later enhanced the nuclear translocation and SUMOylation of p65, enhancing HCC cell proliferation, migration, and consequently disease progression [18].

Essentially, SUMOylation constitutes a network of enzymatic activities that elicit formation of isopeptide bond between the glycine at the C-terminal of a SUMO and the lysine residue of a protein substrate through the mediation of heterodimeric SUMO-activating enzyme (SAE1/SAE2 complex), SUMO-conjugating enzyme UBC9, and SUMO E3 ligases [19,20]. Accruing evidence from numerous studies have suggested that dysregulated *SAE1* expression and/or activity contributes to uncontrolled cell proliferation, development of cancer, angiogenesis, invasion and metastasis [21]. Against the background of reported implication of *SAE1* in SUMOylation and oncogenesis in several malignancies, including glioma and gastric cancer [11,12], and aiming to validate its clinical validity and applicability as a reliable diagnostic and/or prognostic biomarker, the present study explored the expression and of *SAE1*, an indispensable molecular effector of SUMOylation [22], by probing and analyzing clinicopathological data from our in-house HCC cohort and several HCC databases.

In this study, we demonstrated that *SAE1 is* differentially expressed in normal and cancerous tissues, including in paired normal liver and HCC samples. More so, we provided evidence that the enhanced expression of *SAE1* is both grade- and stage-dependent, indicating a probable role for SAE1 in enhanced onco-aggressiveness and disease progression in patients with HCC. This is consistent with reports demonstrating that SUMOylation-dependent transcriptional sub-programming is required for Myc-driven tumorigenesis, and more so implicating *SAE1* in the progression of human glioma and gastric cancer through the activation of SUMOylation-mediated oncogenic signaling pathways [11,12,13].

Additionally, and with clinical relevance, we demonstrated that the overexpression of *SAE1* is associated with poor prognosis, as evident in the shorter overall or disease-specific, and relapse-free survival time of patients with high expression of SAE1 in our HCC cohort and freely accessible larger HCC cohorts. These findings are corroborated by recent report that the expression of key components of the SUMO-involved regulatory network including enhanced *UBE2I* and *SAE1* gene expression levels were strongly linked to poor prognosis in HCC [23] and that the SUMOylation pathway is associated with adverse clinical outcome for patients with multiple myeloma [24].

In addition, we demonstrated that underlying the oncogenic and HCC-promoting activity of *SAE1* was its ability to upregulate oncogenic effectors of cell cycle progression while downregulating FOXO1-associated tumor suppressing signaling. This is consistent with contemporary knowledge that loss of FOXO1 promotes tumor growth and metastasis [25], and accruing evidence that SUMOs such as *SAE1* are essential for the regulation of several cellular processes, including transcriptional regulation, transcript processing, genomic replication and DNA damage repair, where efficiency or inefficiency of the later determines initiation of mitosis or delayed mitotic entry, S-phase arrest, and altered cell cycle progression [26,27,28]; this has significant implication for diseases such as cancer, and suggests that SAE1 is a potential therapeutic target for patients with HCC.

More interestingly, we provided some evidence that SAE1 is a reliable diagnostic biomarker for HCC, with the differential expression of SAE1 in paired normal liver and HCC samples exhibiting an AUC of 0.9252. Similarly, the prognostic relevance of *SAE1* expression was shown with stage dependent AUCs ranging from 0.9091 for stage I to 1.00 for stage IV, and Kaplan-Meier plots indicating worse clinical outcome for patients with high *SAE1* expression compared to their counterparts with low *SAE1* expression. These findings are clinically valid and statistically relevant considering that the ROC curve is a vital tool in disease diagnostics and prognostics, especially where the evaluation of a biomarker’s discriminatory ability is being carried out or for validation of diagnostic and/or prognostic tests. The AUC is the most widely used accuracy index of overall discriminatory power for biomarker identification and validation, such that higher AUC values indicate higher discriminability of a diagnostic or prognostic biomarker or test [29,30].

In conclusion, the present study demonstrates that SAE1 is a targetable SUMO-related molecular biomarker with high potential diagnostic and prognostic implications for patients with HCC.

## Figures and Tables

**Figure 1 cells-10-00178-f001:**
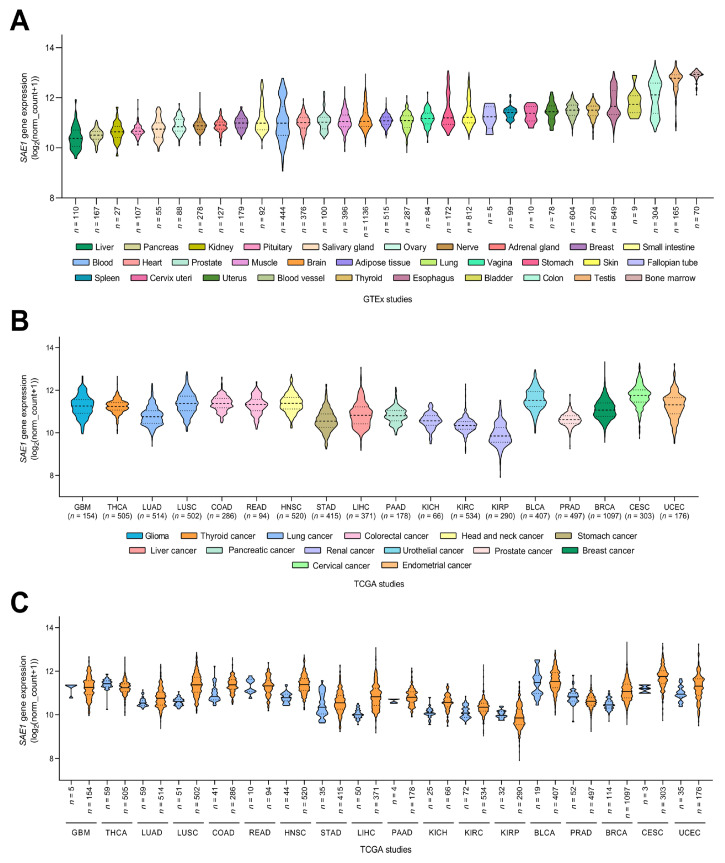
Gene expression profile of SAE1 in Pan-Cancer cohort. Violin plots showing the mRNA expression levels of SAE1 in different human tissue types according to GTEx database (**A**), and in various human cancer types according to TCGA database (**B**). SAE1 expression levels in adjacent tumor (labeled in blue) and tumor samples (labeled in orange) according to TCGA database (**C**).

**Figure 2 cells-10-00178-f002:**
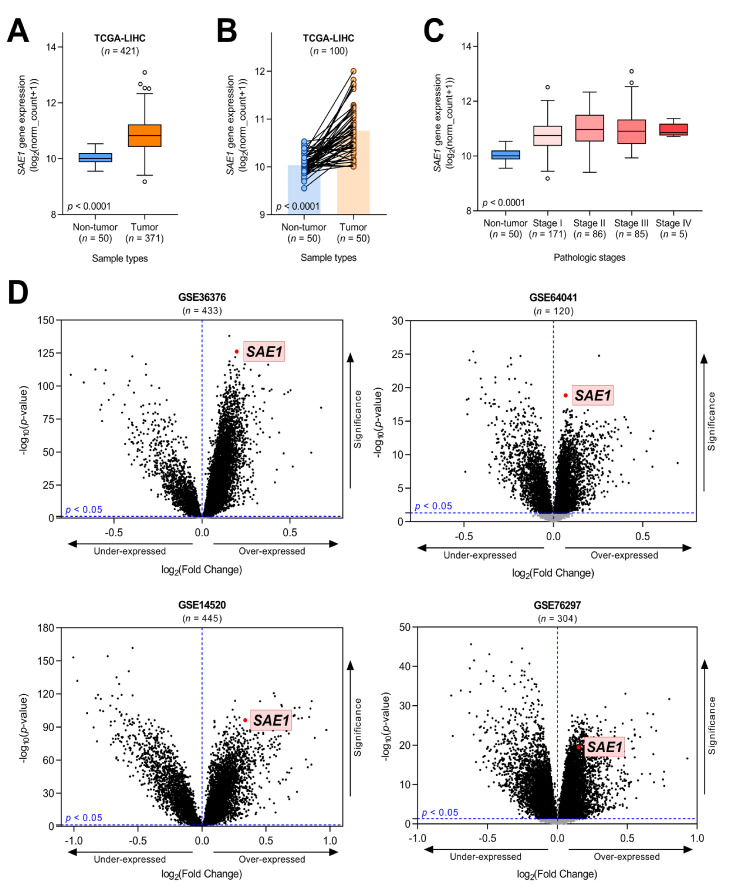
*SAE1* is overexpressed in HCC and associated with disease progression. (**A**) Boxplot showing the mRNA expression levels of *SAE1* in HCC and non-HCC samples according to TCGA−LIHC database (circles are outliers), (**B**) The expression of *SAE1* in the same patients, (**C**) Boxplot showing the *SAE1* expression grouped by pathologic stages, (**D**) Volcano plots indicating *SAE1* upregulated according to GEO databases (GSE36376, GSE64041, GSE14520 and GSE76297).

**Figure 3 cells-10-00178-f003:**
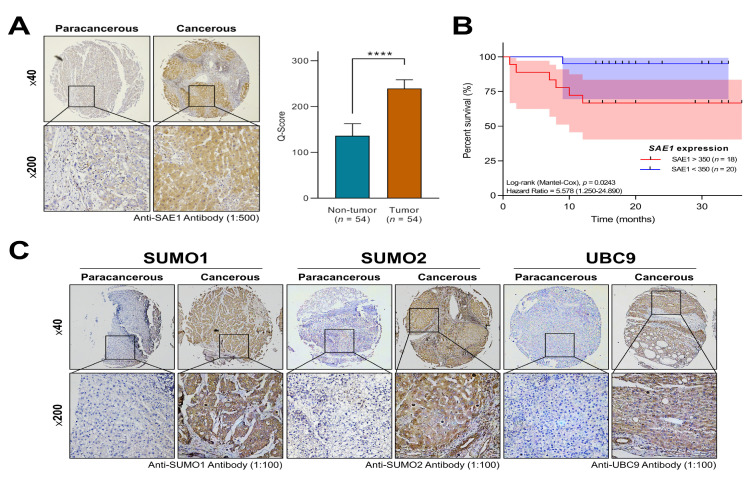
The overexpression of SAE1 is associated with metastasis and poor prognosis in patients with HCC. (**A**) Representative IHC photo-images and graphical representation of SAE1 protein level in paracancerous and cancerous tissues. (**** *p* < 0.0001) (**B**) Kaplan-Meier curve of overall survival according to SAE1protein level of TMU-SHH HCC cohort. (**C**) Representative IHC photo-images of SUMO1, SUMO2, and UBC9 protein levels in paracancerous and cancerous tissues.

**Figure 4 cells-10-00178-f004:**
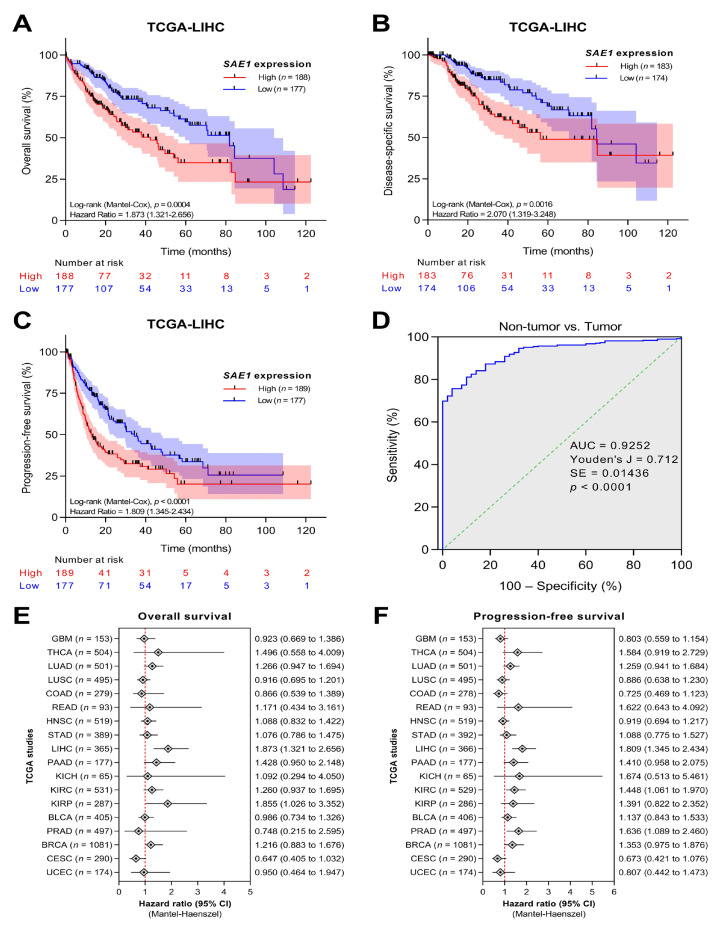
*SAE1* is a reliable diagnostic and prognostic biomarker for HCC. (**A**–**C**) Kaplan–Meier curves of overall survival, disease-specific survival and progression-free interval for HCC patients according to TCGA−LIHC database. (**D**) ROC analysis of *SAE1* expression in non-tumor versus tumor. (**E**,**F**) Forest plot showing hazard ratio estimates and 95% confidence intervals according to TCGA studies.

**Figure 5 cells-10-00178-f005:**
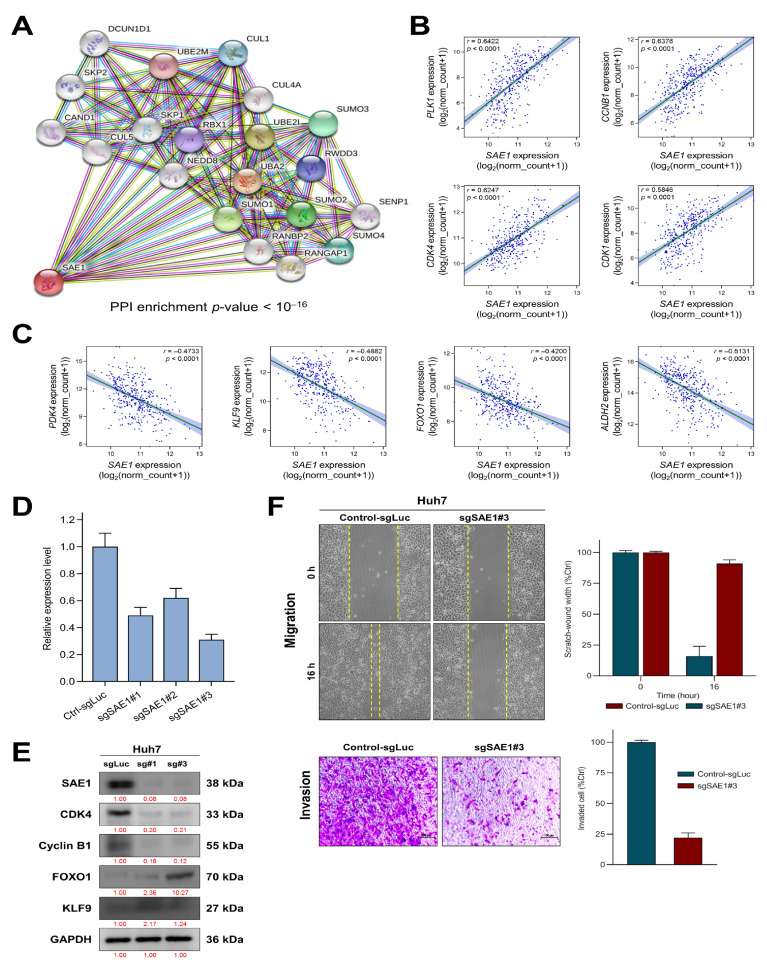
*SAE1* upregulates oncogenic effectors of cell cycle progression while downregulating FOXO1-associated tumor suppressing signaling. (**A**) The SAE1-involved protein–protein interaction network constructed by STRING database. Dots and line plot showing the expression relationship between *SAE1* and (**B**) oncogenes or (**C**) tumor suppressor genes. (**D**) Quantitative real-time PCR analysis validated the CRISPRi knockdown efficacy of sgSAE1s. (**E**) Western blot data of the effect of sgSAE1#1 (sg#1) and sgSAE1#3 (sg#3) on the expression of SAE1, CDK4, cyclin B1, FOXO1, and KLF9 proteins in Huh7 cells. GAPDH served as loading control. (**F**) Representative images of the effect of sg#3 on the migration and invasion of Huh7 cells.

**Figure 6 cells-10-00178-f006:**
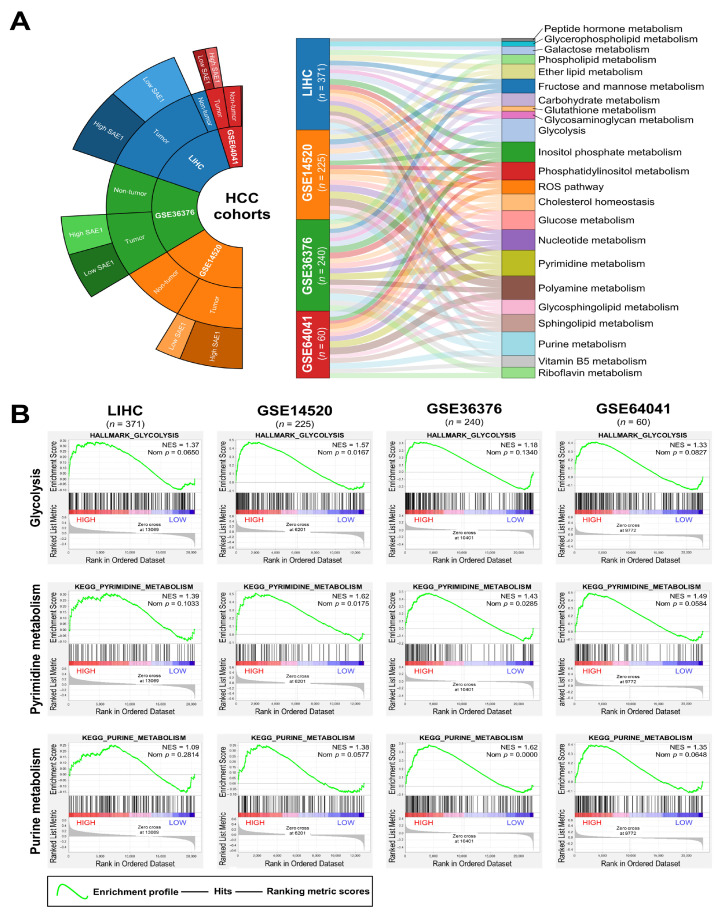
The oncogenic effect of SAE1 is associated with dysregulated cancer metabolic signaling. (**A**) Sankey diagram of the involvement of SAE1 in several metabolic processes in the LIHC, GSE14520, GSE36376, and GSE64041 datasets. (**B**) Representative GSEA plots showing the association between high *SAE1* expression and enriched glycolysis, pyrimidine and purine metabolisms in the LIHC, GSE14520, GSE36376, and GSE64041 datasets.

**Table 1 cells-10-00178-t001:** Patient clinicopathological characteristics of TMU-SHH HCC cohort.

Clinicopathological Variables	Low SAE1 (*n* = 25)		High SAE1 (*n* = 29)		*p*-Value
**Gender** (*n*, %)					
Male	7	28	2	6.9	0.088
Female	18	72	27	93.1	
**Tumor Stage** (*n*, %)					
I + II	18	72	14	48.3	0.021 *
III + IV	7	28	15	51.7	
**Metastasis** (*n*, %)					
M0	18	72	10	48.3	0.036 *
M1	7	28	19	51.7	
**Age** (*n*, %)					
≤65	17	68	16	55.2	0.494
>65	8	32	13	44.8	
**AFP** (*n*, %)					
<400 ng/mL	20	80	10	34.5	0.379
≥400 ng/mL	5	20	19	65.5	
**SAE1** (*n*, %)					
<350	25	100	0	0	<0.001 *
≥350	0	0	29	100	
**Survival Status** (*n*, %)					
Survived	19	76	12	41.4	0.014 *
Expired	2	8	11	37.9	
Lost to follow-up	4	16	6	20.7	

* *p*-value < 0.05.

**Table 2 cells-10-00178-t002:** Univariate and multivariate analysis of SAE1 expression in TMU-SHH cohort.

Clinicopathological Variables	Univariate Analysis			Multivariate Analysis		
	HR	95% CI	*p*-Value	HR	95% CI	*p*-Value
**Gender** Male vs. Female	2.050	0.262–16.017	0.4937	0.575	0.063–5.278	0.6244
**Age, years** ≤65 vs. >65	1.008	0.960–1.057	0.7532	0.968	0.922–1.017	0.1944
**AFP, ng/mL** <400 vs. ≥400	2.375	0.725–7.782	0.1533	1.053	0.290–3.821	0.9378
**Metastasis** M0 vs. M1	10.258	2.206–47.701	0.0030 *	11.500	2.014–65.667	0.0060 *
**SAE1 Q-Score** <350 vs. ≥350	1.026	1.002–1.051	0.0319 *	1.025	1.000–1.049	0.0468 *

* *p*-value < 0.05.

## Data Availability

The datasets used and analyzed in the current study are publicly accessible as indicated in the manuscript.

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
