# Peer review of "SUMO-Activating Enzyme Subunit 1 (SAE1) Is a Promising Diagnostic Cancer Metabolism Biomarker of Hepatocellular Carcinoma"

_cells, 2021, doi:10.3390/cells10010178_

Round 1

Reviewer 1 Report

In this work, authors evidence that SAE1, a SUMO-activating effector of SUMOylation, is overexpressed in HCC samples and its overexpression correlates with poor survival. Authors propose that SAE1 is a targetable biomarker with diagnostic and prognostic value.

This is a bioinformatics study carried out using databases. The only experimental approach taken by authors is immunohistochemistry to prove the overexpression of SAE1 in HCC versus non-tumoral tissue.

The study is interesting but there are some unclear issues (see below) that should be clarified. More importantly, although the study provides nice correlations between SAE1 and clinical variables, it is all hypothetical until authors proved experimentally the pro-oncogenic role of SAE1.

Specific comments:

  1. Table I. In the last clinicopathological variable (survival status) the number of patients does not match with n=25 and n=29 (high SAE1 and low SAE1, respectively).

Additionally, information is somehow confusing:

  • Authors claim that overexpression of SAE1 correlates with poor prognosis and metastasis, but based on data in table I concerning survival, it looks like the survival rate of the high SAE1 group is higher (90.5%) than that (52.2%) of the low SAE1 group.
  • Authors should especify somewhere the meaning of: <350 or >350 for SAE1 (%). Same applies to <400 or >400 for AFP (%).
  • Something must be wrong in this table. I do not understand why all patients in the high SAE1 group belong to the <350 subgroup, instead of the >350; and the 29 patients of the low SAE1 belong to the >350 subgroup.

Likewise, majority of patients of the high SAE1 group (72%) belong to the M0 subgroup in regards to metastasis, and the tumor stage I+II (72%). Please, revise and clarify.

  1. Authors claim to evidence that enhanced expression of SAE1 is stage-dependent. However, graph in figure 2C show similar levels of expression in different tumor stages, at least in stages II-III-IV. Are there significant differences between the subgroups?
  2. The role for SAE1 in promoting disease progression needs to be proved. It would require experiments using HCC cell lines and in vivo models of HCC. Furthermore, authors cannot conclude that SAE1 upregulates oncogenic effectors or downregulates tumor suppressors and interacts with them just based on an expression correlation between SAE1 and those genes (figure 5). Again, specific in vitro experiments using cell lines are needed to prove that SAE1 indeed regulates their expression and/or interacts with these proteins in hepatocarcinoma cells.
  3. Material and Methods. Statistical analysis. Authors mention statistical analysis in cell line experiments, but they have not used cell lines in this study. Please, revise and correct.
  4. It is not clear to me why authors titled the article: “SUMO-activating enzyme subunit 1 (SAE1) is a promising diagnostic cancer metabolism biomarker of hepatocellular carcinoma”. Why cancer metabolism biomarker?? I do not see any specific data regarding cancer metabolism. It is not even discussed.

Minor comments:

  1. Page 4. Lane 136. Abbreviation for liver hepatocellular carcinoma must be included.
  2. Page 9. Lane 206. Disease-specific survival (DSS) should be replaced by disease-free survival (DFS).

Author Response

Point-by-point Response to Reviewers’ Comments:

SUMO-Activating Enzyme Subunit 1 (SAE1) is a Promising Diagnostic Cancer Metabolism Biomarker of Hepatocellular Carcinoma - cells-1044010

Reviewer 1

Q1.1. In this work, authors evidence that SAE1, a SUMO-activating effector of SUMOylation, is overexpressed in HCC samples and its overexpression correlates with poor survival. Authors propose that SAE1 is a targetable biomarker with diagnostic and prognostic value.

This is a bioinformatics study carried out using databases. The only experimental approach taken by authors is immunohistochemistry to prove the overexpression of SAE1 in HCC versus non-tumoral tissue.

The study is interesting but there are some unclear issues (see below) that should be clarified. More importantly, although the study provides nice correlations between SAE1 and clinical variables, it is all hypothetical until authors proved experimentally the pro-oncogenic role of SAE1.

A1.1. We thank the reviewer for carefully reviewing our work and providing insightful comments to help improve our manuscript. We have made necessary revision making use of the suggestions and insights provided.

Q1.2. Specific comments:

  1. Table I. In the last clinicopathological variable (survival status) the number of patients does not match with n=25 and n=29 (high SAE1 and low SAE1, respectively).

A1.2.    We thank the reviewer for drawing our attention to this typographical error. Actually Low SAE1 is n=25, while High SAE1 is n=29. This has now been corrected in the revised manuscript. More so, to address the reviewer’s concern regarding inconsistencies in sample size, we have now also included the number of patients lost to follow-up, as should have been from onset. Please kindly see our revised Table 1 on Page 7, line 233:

Table 1. Patient clinicopathological characteristics of TMU-SHH HCC cohort.

Q1.3. Additionally, information is somehow confusing:

Authors claim that overexpression of SAE1 correlates with poor prognosis and metastasis, but based on data in table I concerning survival, it looks like the survival rate of the high SAE1 group is higher (90.5%) than that (52.2%) of the low SAE1 group.

A1.3. We thank the reviewer for drawing our attention to this typographical error. Actually Low SAE1 is n=25, while High SAE1 is n=29. This has now been corrected in the revised manuscript. More so, to address the reviewer’s concern regarding inconsistencies in sample size, we have now also included the number of patients lost to follow-up, as should have been from onset. Please kindly see our revised Table 1 on Page 7, line 233:

Table 1. Patient clinicopathological characteristics of TMU-SHH HCC cohort.

Q1.4. Authors should especify somewhere the meaning of: <350 or >350 for SAE1 (%). Same applies to <400 or >400 for AFP (%).

A1.4. We thank the reviewer for this comment. As requested, we have included a paragraph stating the meaning of the values in our revised manuscript. Please kindly see our revised text on Page 7, lines 223-232:

3.3. The overexpression of SAE1 is associated with metastasis and poor prognosis in patients with HCC.

The eligible subjects in the TMU-SHH HCC cohort were aged from 25 to 85 with median age of 58.24 for patients with high SAE1 (n = 25) and 61.14 for those with low SAE1 (n = 29). 9 (16.67%) were male and 45 (83.33%) were female. Analyzed clinicopathological data, including patients’ demographic (age and gender) and biochemical profile (AFP, lymph node metastasis, tumor stages and survival status) are summarized in Table 1. The cut-off for AFP is based on clinical consensus that a significantly high level of circulating AFP, greater than 400 ng/mL is suggestive of malignancy of the liver. On the other hand, a threshold cut-off of 350 was ascribed for differential expression of SAE1 based on the quick (Q) score derived from the staining intensity and distribution of SAE1 in the clinical samples.

Q1.5. Something must be wrong in this table. I do not understand why all patients in the high SAE1 group belong to the <350 subgroup, instead of the >350; and the 29 patients of the low SAE1 belong to the >350 subgroup.

Likewise, majority of patients of the high SAE1 group (72%) belong to the M0 subgroup in regard to metastasis, and the tumor stage I+II (72%). Please, revise and clarify.

A1.5. The reviewer is correct. Something was indeed amiss with the initial Table 1. we have now corrected the error and believe things should be clearer now for the reviewer. We sincerely thank the reviewer for drawing our attention to this very embarrassing mistake. As already indicated in A2 and A3, actually Low SAE1 is n=25, while High SAE1 is n=29; thus, the revised values for Low and High SAE1 are <350 and >350, respectively. This has now been corrected in the revised manuscript. This applies for the Metastasis status and Tumor stage, as well. Please kindly see our revised Table 1 on Page 7, line 233:

Table 1. Patient clinicopathological characteristics of TMU-SHH HCC cohort.

Q1.6. Authors claim to evidence that enhanced expression of SAE1 is stage-dependent. However, graph in figure 2C show similar levels of expression in different tumor stages, at least in stages II-III-IV. Are there significant differences between the subgroups?

A1.6. We thank the reviewer for this comment. To address the reviewer’s concern, we have now corrected this semantic issue and rather than “stage-dependent”, we now use “disease progression-associated”. Please kindly see our revised text on Page 6, line 204-209:

Our results indicate that regardless of excluded cases, the median expression of SAE1 mRNA remained significantly upregulated in the tumor samples (~1.1-fold, p < 0.0001) (Figure 2B). Probing for probable role of SAE1 in disease progression, we demonstrated that SAE1 expression increased with HCC stage, as evidenced by higher expression in advanced stages (stages III/IV) and stage II than in stage I or non-tumor (stages II-IV > stage I >> non-tumor) (p < 0.0001), indicating the increased expression of SAE1 is tumorigenic and disease progression-associated (Figure 2C).

Q1.7. The role for SAE1 in promoting disease progression needs to be proved. It would require experiments using HCC cell lines and in vivo models of HCC. Furthermore, authors cannot conclude that SAE1 upregulates oncogenic effectors or downregulates tumor suppressors and interacts with them just based on an expression correlation between SAE1 and those genes (figure 5). Again, specific in vitro experiments using cell lines are needed to prove that SAE1 indeed regulates their expression and/or interacts with these proteins in hepatocarcinoma cells.

A1.7. We appreciate the reviewer’s comments. As requested by the reviewer, we have included some western blot and function assays (migration and invasion) data using the grade III/IV pleomorphic HCC cell line, Huh7 cells. Please kindly see our updated Figure 5 and its legend, Page 12, lines 295-302:

Figure 5. SAE1 upregulates oncogenic effectors of cell cycle progression while downregulating FOXO1-associated tumor suppressing signaling. (A) The SAE1-involved protein-protein interaction network constructed by STRING database. Dots and line plot showing the expression relationship between SAE1 and (B) oncogenes or (C) tumor suppressor genes. (D) Quantitative real-time PCR analysis validated the CRISPRi knockdown efficacy of sgSAE1s.(E) Western blot data of the effect of sgSAE1#1 (sg#1) and sgSAE1#3 (sg#3) on the expression of SAE1, CDK4, cyclin B1, FOXO1, and KLF9 proteins in Huh7 cells. GAPDH served as loading control. (F) Representative images of the effect of sg#3 on the migration and invasion of Huh7 cells.  Please also see our revised Materials & Methods section, Page 3, line 103 - Page 4, line 160:

2.3. Immunohistochemistry

Standard immunohistochemical (IHC) staining and the quantitation of the staining were performed as previously described [15]. Briefly, after de-waxing of the 5μm-thick sections using xylene and re-hydration with ethanol, endogenous peroxidase activity was blocked using 3% hydrogen peroxide. This was followed by antigen retrieval, blocking with 10% normal serum, and incubation of the sections with anti-SAE1 (1:500; #ab185552, Abcam, Cambridge, UK), anti-SUMO1 (1:500; #ab32058, Abcam), anti-SUMO2 (1:500; #ab212838, Abcam), and UBC9 (1:250; #ab75854, Abcam) antibodies overnight at 4°C, followed by goat anti-rabbit IgG (H+L) HRP-conjugated secondary antibody (1:10,000; #65-6120, Thermo Fisher Scientific Inc., Waltham, MA, USA). As chromogenic substrate, Diaminobenzidine (DAB) was used, and the stained sections were counter-stained with Gill’s hematoxylin (Thermo Fisher Scientific, Waltham, MA, USA). The univariate and multivariate analyses were done using the Cox proportional hazards regression model.

2.4. SAE1 knockdown using CRISPR interference

Plasmid vectors containing pLV hU6-sgRNA hUbC-dCas9-KRAB-T2a-Puro (Plasmid #71236) was used for SAE1 knockdown in cells by CRISPR interference (CRISPRi). Three SAE1-specific single-guide RNAs (sgRNAs) designed using the online tool CHOPCHOP (http://chopchop.cbu.uib.no) were synthesized and separately cloned into lenti-dCas9-KRAB. Lentiviruses were packaged and transfected into Huh7 cells. Transfected monoclonal Huh7 cells were selected by 2 μg/mL puromycin. The cell construction with knockdown of SAE1 was verified by genomic sequencing and quantitative real-time PCR. The sgRNA sequences for SAE1 are as follows: sgSAE1#1 (sg#1) 5’-GTGCCACATAAGTGACCACG-3’, sgSAE1#2 (sg#2) 5’-GGCGACTGCATGTCACGTGA-3’ and  sgSAE1#3 (sg#3) 5’-ACGAGGTACT GCGCAGGCGT-3’.

2.5. Real-time PCR reaction

Quantitative real-time PCR reaction was performed as previous described in [15] using the following primers: SAE1-FP: 5’-AGGACTGACCATGCTGGATCAC-3’ and SAE1-RP: 5’-CTCAGTGTCC ACCTTCACATCC-3’.

2.6. Western blot analysis

Total protein lysate was prepared from cultured HCC cells using ice-cold lysis buffer solution. After boiling at 95°C for 5 min, immunobloting was performed. Blots were blocked with 5% non-fat milk in Tris Buffered Saline with Tween 20 (TBST) for 1 h, incubated at 4°C overnight with specific primary antibodies against SAE1 (1:1000; #13585S, Cell Signaling Technology, Inc., Danvers, MA, USA), CDK4 (1:1000; #2906, Cell Signaling Technology), Cyclin B1 (1:500; Sc-245, Santa Cruz Biotechnology, Dallas, TX, USA), FOXO1 (1:1000; #2880, Cell Signaling Technology), GAPDH (1:500; Sc-47724, Santa Cruz Biotechnology), and KLF9 (1:1000; ab227920, Abcam, Cambridge Inc., UK) in Supplementary Table S1. Thereafter, the polyvinylidene difluoride (PVDF) membranes were washed thrice with TBST, incubated with horseradish peroxidase (HRP)-labeled secondary antibody for 1 h at room temperature and then washed with TBST again before band detection using enhanced chemiluminescence (ECL) Western blotting reagents and imaging with the BioSpectrum Imaging System (UVP, Upland, CA, USA).

2.7. Transwell matrigel invasion assay

After pre-coating the chamber membranes (8 μm, BD Falcon) with Bmatrigel at 4°C overnight, the wild type (WT) or CRISPRi SAE1-knockdown cells were seeded at a density of 1×105 cells per chamber. DMEM with 1% fetal bovine serum (FBS) supplement was added to the upper chamber and DMEM containing 10% FBS added to the lower chamber. Cells were incubated for 48 h. The non-invading cells on the top of membranes was carefully removed using sterile cotton swab, and the invaded cells that penetrate the membrane were fixed in ethanol, followed by crystal violet staining. The number of invaded cells was counted under the microscope in five random fields of vision and representative images photographed.

2.8. Scratch-wound migration assay

Cell migration potential was evaluated using the wound healing assay. Briefly, wild type (WT) or CRISRi SAE1-knockdown Huh7 cells were seeded onto 6-well plates (Corning Inc., Corning, NY, USA) with complete growth media containing 10% FBS, and cultured till 99–100% confluence was attained. The cell monolayers were scratched with sterile yellow pipette tip along the median axes of the culture wells. The cell migration images were captured at the 0 and 16 h time points after denudation, under a microscope with a 10× objective lens, and analyzed with the NIH ImageJ software (https://imagej.nih.gov/ij/download.html).

Also kindly see revised text, Page 12, lines 303-318:

Furthermore, we used the cBioPortal for Cancer Genomics (https://www.cbioportal.org/) for the identification of genes with significant positive or negative correlation with SAE1 in the TCGA-LIHC cohort. Our results showed that SAE1 is strongly co-expressed with the cell cycle-related oncogenes PLK1 (r = 0.64, p < 0.0001), CCNB1 (r = 0.64, p < 0.0001), CDK4 (r = 0.58, p < 0.0001) and CDK1 (r = 0.58, p < 0.0001) (Figure 5B), but inversely related to tumor suppressor genes PDK4 (r = -0.47, p < 0.0001), KLF9 (r = -0.47, p < 0.0001), FOXO1 (r = -0.42, p < 0.0001) and ALDH2 (r = -0.5131, p < 0.0001) (Figure 5C). To gain inside into the significance of SAE1 in hepatocarcinogenesis, we knocked-down the gene using CRISPRi and validated the knockdown efficacy by real-time PCR analysis. As shown in Figure 5D, sgSAE1#1 and sgSAE1#3 exhibited high knockdown efficacy with roughly 50% and 30%, respectively. Consistent with these data, results of our western blot analysis show that silencing SAE1, elicited upregulated expression of drivers of cancer progression CDK4 and cyclin B1 (CCNB1 gene product), concomitantly with downregulated tumor suppressors FOXO1 and KLF9 proteins in HCC cell line Huh7 (Figure 5E). Similarly, sgSAE1#3 significantly suppressed the ability of the Huh7 cells to invade (5.2-fold, p < 0.001) or migrate (6.1-fold, p < 0.001) (Figure 5F). These findings suggest that SAE1 upregulates oncogenic effectors of cell cycle progression while downregulating FOXO1-associated tumor suppressing signaling.

Q1.8. Material and Methods. Statistical analysis. Authors mention statistical analysis in cell line experiments, but they have not used cell lines in this study. Please, revise and correct.

A1.8. We thank the reviewer for this comment. Cell line experiments are now included in the reised manuscript and the line regarding cell line in the Statistical analysis section is consistent with the included data.

Q1.9. It is not clear to me why authors titled the article: “SUMO-activating enzyme subunit 1 (SAE1) is a promising diagnostic cancer metabolism biomarker of hepatocellular carcinoma”. Why cancer metabolism biomarker?? I do not see any specific data regarding cancer metabolism. It is not even discussed.

A1.9. We sincerely thank the reviewer for this comment. We have now included our data on the association of SAE1 to stress signaling, especially a dysregulated metabolic landscape in the revise manuscript. Please kindly see the revised title on Title page, Page 13, lines 319-332:

3.6. The oncogenic effect of upregulated SAE1 is associated with dysregulated cancer metabolic signaling

Understanding that several stress signals, including hypoxia, impaired metabolism, nutrient deficiency, DNA damage (genotoxic stress) and dysregulated nucleotide metabolism, facilitate the initiation and development of cancer, and that dysregulated SUMOylation can play a crucial role in the protection of cancer cells from exogenous or endogenous stress signals [21], we investigated likely association of SAE1 expression with hypoxia and impaired metabolism. Results of our gene set enrichment analysis of the LIHC (n = 371), GSE14520 (n = 225), GSE36376 (n = 240), and GSE64041 (n = 60) HCC datasets showed the existence of significant positive correlation between high SAE1 and dysregulated reactive oxygen species (ROS), glycolysis, and cholesterol homeostasis pathways in patients with HCC (Figure 6A). Gene Set Enrichment Analysis (GSEA) plots using HCC datasets implied the upregulated co-expression of SAE1 and biomarkers of glucose metabolism, pyrimidine metabolism, and purine metabolism (Figure 6B). These data do indicate, at least in part, that the oncogenic effect of upregulated SAE1 is associated with dysregulated cancer metabolic signaling.

Also kindly see the newly included Figure 6 with its legend, Page 14, lines 335-339:

Figure 6. The oncogenic effect of SAE1 is associated with dysregulated metabolic signalings. (A) Sankey diagram of the involvement of SAE1 in several metabolic processes in the LIHC, GSE14520, GSE36376, and GSE64041 datasets. (B) Representative GSEA plots showing the association between high SAE1 expression and enriched glycolysis, pyrimidine and purine metabolisms in the LIHC, GSE14520, GSE36376, and GSE64041 datasets.

Q1.10. Minor comments:

  1. Page 4. Lane 136. Abbreviation for liver hepatocellular carcinoma must be included.
  2. Page 9. Lane 206. Disease-specific survival (DSS) should be replaced by disease-free survival (DFS).

A1.10. We thank the reviewer for the comment. We have now included the abbreviation for liver hepatocellular carcinoma in our revised paper as requested by the reviewer. Please kindly see Page 5, lines 184-187:

Further exploring the SAE1 mRNA levels in paired tumor−non-tumor samples from patients with one of 18 different cancer types using the Cancer Genome Atlas (TCGA) datasets, we observed that SAE1 was significantly more expressed in liver hepatocellular carcinoma (LIHC, n = 371), compared to their normal tissue counterparts (n = 50) (Figures 1B).

Secondly, we have corrected the legend for Figure 4B. Please kindly see our Page 10, lines 267-271.

Figure 4. SAE1 is a reliable diagnostic and prognostic biomarker for HCC. (A-C) Kaplan-Meier curves of overall survival, disease-free survival and progression-free interval for HCC patients according to TCGA LIHC database. (D) ROC analysis of SAE1 expression in non-tumor versus tumor. (E-F) Forest plot showing hazard ratio estimates and 95% confidence intervals according to TCGA studies.

Reviewer 2 Report

Cells (ISSN 2073-4409)

Manuscript ID; cells-1044010

SUMO-Activating Enzyme Subunit 1 (SAE1) is a Promising Diagnostic Cancer Metabolism Biomarker of Hepatocellular Carcinoma

Jiann Ruey Ong , Oluwaseun Adebayo Bamodu , Nguyen Viet Khang , Yen-Kuang Lin , Chi-Tai Yeh , Wei-Hwa Lee , Yih-Giun Cherng.

Abstract:

Background: Hepatocellular carcinoma (HCC) is one of the most diagnosed malignancies and a leading cause of cancer-related mortality globally. This is exacerbated by its highly aggressive phenotype, and limitation in early diagnosis and effective therapies. The SUMO-activating enzyme subunit 1 (SAE1) is a component of a heterodimeric small ubiquitin-related modifier that plays a vital role in SUMOylation, a post-translational modification involving in cellular events such as regulation of transcription, cell cycle and apoptosis. Reported overexpression of SAE1 in glioma in a stage-dependent manner suggests it probable role in cancer initiation and progression.

Methods: In this study, hypothesizing that SAE1 is implicated in HCC metastatic phenotype and poor prognosis, we analyzed the expression of SAE1 in several cancer databases.

Results: Here, we demonstrated that SAE1 is overexpressed in HCC samples compared to normal liver tissue, and this observed SAE1 overexpression is stage and grade-dependent and associated with poor survival. The receiver operating characteristic analysis of SAE1 in TCGA-LIHC patients (n=421) showed an AUC of 0.925, indicating an excellent diagnostic value of SAE1 in HCC. Our protein-protein interaction analysis for SAE1 showed that SAE1 interacted with and activated oncogenes such as PLK1, CCNB1, CDK4 and CDK1, while simultaneously inhibiting tumor suppressors including PDK4, KLF9, FOXO1 and NEDD4. Immunohistochemical staining and clinicopathological correlate analysis of SAE1 in our TMU-SHH HCC cohort (n =54) further validated the overexpression of SAE1 in cancerous liver tissues compared with ‘normal’ paracancerous tissue, and high SAE1 expression was strongly correlated with metastasis and disease progression.

Conclusion: In conclusion, the present study demonstrates that SAE1 is a targetable molecular biomarker with high potential diagnostic and prognostic implications for patients with HCC.

--------------------------------------------------------------------------

It is a topic of interest to the researchers in the related area but the paper needs minor improvements before acceptance for publication. My detailed comments are as follows:

  1. The introduction, materials and methods in the paper works very well, especially the part that correspond to data acquisition and statistical analysis. In my opinion it would include more references regarding sumoylation in HCC. At present, new targets are being explored, not only in cancer in general, but also in HCC, which implies the relevant role of PTMs (sumoylation, ubiquitination, needylation) in the progression of the disease. Figure 5a, of this good work, is a reflection of the interaction of different ub-likes. SUMO1, 2, 3 (and 4), NEDD8 and some cullins, SENPs. There are recent works that already analyze the importance of UBC9 and SUMO in HCC. I consider that it would be important to introduce new references that support the very good results of this work.

  • SUMOylation regulates LKB1 localization and its oncogenic activity in liver cancer. EBioMedicine. 2019 Feb;40:406-421. doi: 10.1016/j.ebiom.2018.12.031. Epub 2018 Dec 26. PMID: 30594553; PMCID: PMC6412020.

  • Small ubiquitin-related modifier 1 is involved in hepatocellular carcinoma progression via mediating p65 nuclear translocation. 2016 Apr 19;7(16):22206-18. doi: 10.18632/oncotarget.8066. PMID: 26993772; PMCID: PMC5008356.

  • S-adenosyl methionine regulates ubiquitin-conjugating enzyme 9 protein expression and sumoylation in murine liver and human cancers. Hepatology. 2012 Sep;56(3):982-93. doi: 10.1002/hep.25701. Epub 2012 Jul 12. PMID: 22407595; PMCID: PMC3378793.

  1. As I mentioned earlier, the materials and methods and the results are well structured, however there are data that in my opinion are lost. Sumoylation is a process that takes place in three consecutive steps. The first step involves activating SUMO (SAE1). The congugation, in which UBC9, plays a fundamental and unique role. Finally, the possible action of a ligase (E3) provides added value to sumoylation. In this work in figure 3A, only one immunohistochemistry with SAE1 is presented. I think it would be very important to increase confidence in the very clear results obtained, to include immunohistochemistry with UBC9 and SUMO (sumo1 and 2 at least).

  1. I consider, once again as I have mentioned previously, that a few minimal interventions and contributions are necessary to increase the quality (already good) of the work.

Author Response

Reviewer 2

Q2.1. It is a topic of interest to the researchers in the related area but the paper needs minor improvements before acceptance for publication. My detailed comments are as follows:

A2.1. We thank the reviewer for carefully reviewing our work and providing insightful comments to help improve our manuscript. We have made necessary revision making use of the suggestions and insights provided.

Q2.2. The introduction, materials and methods in the paper works very well, especially the part that correspond to data acquisition and statistical analysis. In my opinion it would include more references regarding sumoylation in HCC. At present, new targets are being explored, not only in cancer in general, but also in HCC, which implies the relevant role of PTMs (sumoylation, ubiquitination, needylation) in the progression of the disease. Figure 5a, of this good work, is a reflection of the interaction of different ub-likes. SUMO1, 2, 3 (and 4), NEDD8 and some cullins, SENPs. There are recent works that already analyze the importance of UBC9 and SUMO in HCC. I consider that it would be important to introduce new references that support the very good results of this work.

  • SUMOylation regulates LKB1 localization and its oncogenic activity in liver cancer. EBioMedicine. 2019 Feb;40:406-421. doi: 10.1016/j.ebiom.2018.12.031. Epub 2018 Dec 26. PMID: 30594553; PMCID: PMC6412020.
  • Small ubiquitin-related modifier 1 is involved in hepatocellular carcinoma progression via mediating p65 nuclear translocation. 2016 Apr 19;7(16):22206-18. doi: 10.18632/oncotarget.8066. PMID: 26993772; PMCID: PMC5008356.
  • S-adenosyl methionine regulates ubiquitin-conjugating enzyme 9 protein expression and sumoylation in murine liver and human cancers. Hepatology. 2012 Sep;56(3):982-93. doi: 10.1002/hep.25701. Epub 2012 Jul 12. PMID: 22407595; PMCID: PMC3378793.

 A2.2. We thank the reviewer for this suggestion. As suggested, we have now included more references regarding SUMOylation, as well as those provided by the reviewer. Please kindly see our revised Discussion section, Page 15, lines 341-364:

  1. Discussion

Hepatocarcinogenesis entails alteration of cellular gene expression with consequent loss of benignity, acquisition of malignant phenotype and enhancement of the aggressiveness of the resultant cancerous liver cells [3]. Previous studies have shown that post-translational modification such as ubiquitination and SUMOylation which play very crucial roles in the non-static regulation of protein structure, stability, intracellular localization, activity, function, and interaction with other proteins, are significantly enhanced in HCC [14, 17]. Recently, it was reported that the SUMO2-mediated SUMOylation of the supposedly tumor suppressor, liver kinase B1 (LKB1), facilitated hepatocarcinogenesis and disease progression in in vivo mice models and human HCC cohort, especially in hypoxic conditions [17]. Consistent with this report, it has also been shown that hypoxia or exposure to TNF-α upregulated SUMO1 expression and the later enhanced the nuclear translocation and SUMOylation of p65, enhancing HCC cell proliferation, migration, and consequently disease progression [18].

Essentially, SUMOylation constitutes a network of enzymatic activities that elicit formation of isopeptide bond between the glycine at the C-terminal of a SUMO and the lysine residue of a protein substrate through the mediation of heterodimeric SUMO-activating enzyme (SAE1/SAE2 complex), SUMO-conjugating enzyme UBC9, and SUMO E3 ligases [19, 20]. Accruing evidence from numerous studies have suggested that dysregulated SAE1 expression and/or activity contributes to uncontrolled cell proliferation, development of cancer, angiogenesis, invasion and metastasis [21]. Against the background of reported implication of SAE1 in SUMOylation and oncogenesis in several malignancies, including glioma and gastric cancer [11, 12], and aiming to validate its clinical validity and applicability as a reliable diagnostic and/or prognostic biomarker, the present study explored the expression and of SAE1, an indispensable molecular effector of SUMOylation [22], by probing and analyzing clinicopathological data from our in-house HCC cohort and several HCC databases.

Please also kindly see our updated Reference section, Page 18, lines 472-487:

Zubiete-Franco I, García-Rodríguez JL, Lopitz-Otsoa F, Lopitz-Otsoa F, Serrano-Macia M, Simon J, et al. SUMOylation regulates LKB1 localization and its oncogenic activity in liver cancer. EBioMedicine. 2019; 40:406-421.

Liu J, Tao X, Zhang J, Wang P, Sha M, Ma Y, Geng X, Feng L, Shen Y, Yu Y, Wang S, Fang S, Shen Y. Small ubiquitin-related modifier 1 is involved in hepatocellular carcinoma progression via mediating p65 nuclear translocation. Oncotarget. 2016 Apr 19;7(16):22206-18.

Pichler A, Fatouros C, Lee H, Eisenhardt N. SUMO conjugation – a mechanistic view. Biomolecular Concepts. 2017; 8(1): 13-36.

Tomasi ML, Tomasi I, Ramani K, Pascale RM, Xu J, Giordano P, Mato JM, Lu SC. S-adenosyl methionine regulates ubiquitin-conjugating enzyme 9 protein expression and sumoylation in murine liver and human cancers. Hepatology. 2012 Sep;56(3):982-93.

Han ZJ, Feng YH, Gu BH, Li YM, Chen H. The post-translational modification, SUMOylation, and cancer (Review). Int J Oncol. 2018; 52(4): 1081-1094.

Wu R, Fang J, Liu M, A J, Liu J, Chen W, et al. SUMOylation of the transcription factor ZFHX3 at Lys-2806 requires SAE1, UBC9, and PIAS2 and enhances its stability and function in cell proliferation. J Biol Chem. 2020; 295(19):6741-6753.

Q2.3. As I mentioned earlier, the materials and methods and the results are well structured, however there are data that in my opinion are lost. Sumoylation is a process that takes place in three consecutive steps. The first step involves activating SUMO (SAE1). The congugation, in which UBC9, plays a fundamental and unique role. Finally, the possible action of a ligase (E3) provides added value to sumoylation. In this work in figure 3A, only one immunohistochemistry with SAE1 is presented. I think it would be very important to increase confidence in the very clear results obtained, to include immunohistochemistry with UBC9 and SUMO (sumo1 and 2 at least).

A2.3. We sincerely thank the reviewer for this insightful comment. As suggested by the reviewer we have now included IHC staining results for UBC9, SUMO-1 and SUMO-2.  Please kindly see our revised result section, Page 8, line 241-252:

Corroborating the findings from big data analysis, IHC staining of samples from our TMU-SHH HCC cohort (n = 54) showed a 1.85-fold upregulated expression of SAE protein in the cancerous HCC compared to the non-tumor para-cancer liver tissue (p < 0.0001) (Figure 3A).

Probing for clinical relevance of the observed high expression of SAE1 protein, using the Kaplan-Meier curve for survival analysis, we demonstrated that compared to patients with low SAE1 expression (n = 20), those with high SAE1 expression (n = 18) exhibited worse overall survival ((HR (95%CI): 5.578 (1.250-24.890); p = 0.024)) (Figure 3B). Consistent with the vital role of SUMO proteins and the UBC9 in sumoylation [18-21], we further demonstrated that similar to SAE1, the expression levels of SUMO1, SUMO2, and UBC9 were upregulated in the HCC tissues compared to their non-tumor paracancerous counterparts (Figure 3C). These results indicate that overexpression of SAE1, a critical component of the sumoylation complex, is associated with metastasis and poor prognosis in patients with HCC.

Please kindly see our updated Figure 3 and its legend, Page 9, line 254-258:

Figure 3. The overexpression of SAE1 is associated with metastasis and poor prognosis in patients with HCC. (A) Representative IHC photo-images and graphical representation of SAE1 protein level in paracancerous and cancerous tissues. (B) Kaplan-Meier curve of overall survival according to SAE1protein level of TMU-SHH HCC cohort. (C) Representative IHC photo-images of SUMO1, SUMO2, and UBC9 protein levels in paracancerous and cancerous tissues.

Please also see our revised Materials & Methods section, Page 3, line 103 - Page 4, line 160:

2.3. Immunohistochemistry

Standard immunohistochemical (IHC) staining and the quantitation of the staining were performed as previously described [15]. Briefly, after de-waxing of the 5μm-thick sections using xylene and re-hydration with ethanol, endogenous peroxidase activity was blocked using 3% hydrogen peroxide. This was followed by antigen retrieval, blocking with 10% normal serum, and incubation of the sections with anti-SAE1 (1:500; #ab185552, Abcam, Cambridge, UK), anti-SUMO1 (1:500; #ab32058, Abcam), anti-SUMO2 (1:500; #ab212838, Abcam), and UBC9 (1:250; #ab75854, Abcam) antibodies overnight at 4°C, followed by goat anti-rabbit IgG (H+L) HRP-conjugated secondary antibody (1:10,000; #65-6120, Thermo Fisher Scientific Inc., Waltham, MA, USA). As chromogenic substrate, Diaminobenzidine (DAB) was used, and the stained sections were counter-stained with Gill’s hematoxylin (Thermo Fisher Scientific, Waltham, MA, USA). The univariate and multivariate analyses were done using the Cox proportional hazards regression model.

2.4. SAE1 knockdown using CRISPR interference

Plasmid vectors containing pLV hU6-sgRNA hUbC-dCas9-KRAB-T2a-Puro (Plasmid #71236) was used for SAE1 knockdown in cells by CRISPR interference (CRISPRi). Three SAE1-specific single-guide RNAs (sgRNAs) designed using the online tool CHOPCHOP (http://chopchop.cbu.uib.no) were synthesized and separately cloned into lenti-dCas9-KRAB. Lentiviruses were packaged and transfected into Huh7 cells. Transfected monoclonal Huh7 cells were selected by 2 μg/mL puromycin. The cell construction with knockdown of SAE1 was verified by genomic sequencing and quantitative real-time PCR. The sgRNA sequences for SAE1 are as follows: sgSAE1#1 (sg#1) 5’-GTGCCACATAAGTGACCACG-3’, sgSAE1#2 (sg#2) 5’-GGCGACTGCATGTCACGTGA-3’ and sgSAE1#3 (sg#3) 5’-ACGAGGTACT GCGCAGGCGT-3’.

2.5. Real-time PCR reaction

Quantitative real-time PCR reaction was performed as previous described in [15] using the following primers: SAE1-FP: 5’-AGGACTGACCATGCTGGATCAC-3’ and SAE1-RP: 5’-CTCAGTGTCC ACCTTCACATCC-3’.

2.6. Western blot analysis

Total protein lysate was prepared from cultured HCC cells using ice-cold lysis buffer solution. After boiling at 95°C for 5 min, immunobloting was performed. Blots were blocked with 5% non-fat milk in Tris Buffered Saline with Tween 20 (TBST) for 1 h, incubated at 4°C overnight with specific primary antibodies against SAE1 (1:1000; #13585S, Cell Signaling Technology, Inc., Danvers, MA, USA), CDK4 (1:1000; #2906, Cell Signaling Technology), Cyclin B1 (1:500; Sc-245, Santa Cruz Biotechnology, Dallas, TX, USA), FOXO1 (1:1000; #2880, Cell Signaling Technology), GAPDH (1:500; Sc-47724, Santa Cruz Biotechnology), and KLF9 (1:1000; ab227920, Abcam, Cambridge Inc., UK) in Supplementary Table S1. Thereafter, the polyvinylidene difluoride (PVDF) membranes were washed thrice with TBST, incubated with horseradish peroxidase (HRP)-labeled secondary antibody for 1 h at room temperature and then washed with TBST again before band detection using enhanced chemiluminescence (ECL) Western blotting reagents and imaging with the BioSpectrum Imaging System (UVP, Upland, CA, USA).

2.7. Transwell matrigel invasion assay

After pre-coating the chamber membranes (8 μm, BD Falcon) with Bmatrigel at 4°C overnight, the wild type (WT) or CRISPRi SAE1-knockdown cells were seeded at a density of 1×105 cells per chamber. DMEM with 1% fetal bovine serum (FBS) supplement was added to the upper chamber and DMEM containing 10% FBS added to the lower chamber. Cells were incubated for 48 h. The non-invading cells on the top of membranes was carefully removed using sterile cotton swab, and the invaded cells that penetrate the membrane were fixed in ethanol, followed by crystal violet staining. The number of invaded cells was counted under the microscope in five random fields of vision and representative images photographed.

2.8. Scratch-wound migration assay

Cell migration potential was evaluated using the wound healing assay. Briefly, wild type (WT) or CRISRi SAE1-knockdown Huh7 cells were seeded onto 6-well plates (Corning Inc., Corning, NY, USA) with complete growth media containing 10% FBS, and cultured till 99–100% confluence was attained. The cell monolayers were scratched with sterile yellow pipette tip along the median axes of the culture wells. The cell migration images were captured at the 0 and 16 h time points after denudation, under a microscope with a 10× objective lens, and analyzed with the NIH ImageJ software (https://imagej.nih.gov/ij/download.html).

Q2.4. I consider, once again as I have mentioned previously, that a few minimal interventions and contributions are necessary to increase the quality (already good) of the work.

A2.4. Once again, we sincerely thank the reviewer for carefully reviewing our work and providing insightful comments to help improve the quality of our manuscript. We have made necessary revision making use of the suggestions and insights provided.

Round 2

Reviewer 1 Report

Authors have addressed all issues raised during peer review (round 1). I would like to recognize authors´ effort and work to run functional in vitro assays in such a short period of time. I consider that the changes done have improved the quality of the manuscript.

I just have a specific comment about the in vitro wound healing assay (scratch wound migration assay) performed by the authors: 

based on the description provided in methods section, it seems that authors have used DMEM supplemented with 10% FBS during the assay. However, this is not optimal because of the interference of a cell proliferation response with the migration response, which compromise results interpretation. To prevent such interference after scratching complete medium should be replaced by either DMEM not supplemented with serum (or supplemented with a very low % of serum if cells would not tolerate complete starvation) or alternatively low concentrations of mitomycin C can be added for a few hours before scratching to inhibit DNA synthesis. Authors should clarify this issue, whether or not serum was maintained during the whole assay and if that is the case how it is possible to discern between proliferation and migration responses. 

Author Response

Point-by-point Response to Reviewers’ Comments:

SUMO-Activating Enzyme Subunit 1 (SAE1) is a Promising Diagnostic Cancer Metabolism Biomarker of Hepatocellular Carcinoma - cells-1044010

Reviewer 1

Q1.1. Authors have addressed all issues raised during peer review (round 1). I would like to recognize authors´ effort and work to run functional in vitro assays in such a short period of time. I consider that the changes done have improved the quality of the manuscript.

A1.1. We thank the reviewer for carefully reviewing our work and providing insightful comments to help improve our manuscript. We have made necessary revision making use of the suggestions and insights provided.

Q1.2. I just have a specific comment about the in vitro wound healing assay (scratch wound migration assay) performed by the authors: based on the description provided in methods section, it seems that authors have used DMEM supplemented with 10% FBS during the assay. However, this is not optimal because of the interference of a cell proliferation response with the migration response, which compromise results interpretation. To prevent such interference after scratching complete medium should be replaced by either DMEM not supplemented with serum (or supplemented with a very low % of serum if cells would not tolerate complete starvation) or alternatively low concentrations of mitomycin C can be added for a few hours before scratching to inhibit DNA synthesis. Authors should clarify this issue, whether or not serum was maintained during the whole assay and if that is the case how it is possible to discern between proliferation and migration responses.

A1.2. We sincerely thank the reviewer for pointing out this embarrassing typographical error. We actually used and to write intended 0.2% and not 10%.  Thus, we have now corrected the mistake in our revised version. Please kindly see our revised Materials & Methods section, Page 4, Lines 151 -159:

2.8. Scratch-wound migration assay

Cell migration potential was evaluated using the wound healing assay. Briefly, wild type (WT) or CRISRi SAE1-knockdown Huh7 cells were seeded onto 6-well plates (1 × 106 cells/well) (Corning Inc., Corning, NY, USA) with complete growth media containing 0.2% FBS, and cultured till 95–100% confluence was attained. That would help cells not going to apoptosis or necrosis, but also no proliferation occurs. The cell monolayers were scratched with sterile yellow pipette tip along the median axes of the culture wells. The cell migration images were captured at the 0 and 16 h time points after denudation, under a microscope with a 10× objective lens, and analyzed with the NIH ImageJ software (https://imagej.nih.gov/ij/download.html).